# Extraperitoneal Laparoscopic Prostatectomy in a Prostate Cancer Patient Undergoing Continuous Ambulatory Peritoneal Dialysis

**DOI:** 10.3390/life12030413

**Published:** 2022-03-12

**Authors:** Damian Widz, Iga Kuliniec, Andriy Yadlos, Damian Sudoł, Michał Godzisz, Agata Wisz, Paweł Płaza, Przemysław Mitura, Michał Cabanek, Krzysztof Bar

**Affiliations:** 1Department of Urology and Oncological Urology, Medical University of Lublin, Jaczewskiego 8, 20-954 Lublin, Poland; damian.widz@umlub.pl (D.W.); 61177@student.umlub.pl (A.Y.); 45456@student.umlub.pl (D.S.); 45392@student.umlub.pl (M.G.); pawel.plaza@umlub.pl (P.P.); przemyslaw.mitura@umlub.pl (P.M.); oddzial.urologii@spsk4.lublin.pl (M.C.); krzysztof.bar@umlub.pl (K.B.); 2Department of Diagnostic Imaging, Radiology and Nuclear Medicine, Faculty of Medical Science in Katowice, Medical University of Silesia, Medyków 16, 40-752 Katowice, Poland; d201024@365.sum.edu.pl

**Keywords:** radical prostatectomy, prostate cancer, CAPD, extraperitoneal laparoscopy

## Abstract

Radical prostatectomy (RP) in patients on continuous ambulatory peritoneal dialysis (CAPD) is a challenging procedure. The following key points need to be considered: the peritoneal cavity integrity, adjustment of the trocar positions to the peritoneal dialysis (PD) tube location, and the oncological and functional outcomes. We present a clinical case of a patient on CAPD, incidentally diagnosed with prostate cancer (PCa) during the pre-transplant evaluation. The patient suffered from LUTS, due to bladder outlet obstruction (BOO). A transurethral bladder neck incision (TUNI), with median lobe resection, was performed. A PCa Gleason score of six (3 + 3) was found in the histopathological specimen. The primary procedure was complicated by bladder neck sclerosis and acute urinary retention (AUR), resolved by suprapubic cystostomy. After proper staging determination, the patient was qualified for laparoscopic extraperitoneal RP. The standard trocar placement was modified to align with the route of the PD tube, and Retzius’ space scarring was released to allow extraperitoneal prostatectomy. There were no signs of peritoneal wall damage or dialysis tube displacement. Peritoneal dialysis was resumed after 4 weeks. Laparoscopic extraperitoneal RP should be considered as an acceptable treatment method for selected patients with localized prostate cancer, allowing CAPD resumption. To the best of our knowledge, this is the first report of retroperitoneal laparoscopic RP being used in the PD population.

## 1. Introduction

According to Global Cancer Statistics 2020, prostate cancer (PCa) is the second most diagnosed cancer among men worldwide. It is also the fifth leading cause of cancer deaths among men [1]. In 2017, prostate cancer became the most frequently occurring cancer among Polish men [2]. With an increasing number of urologic patients who often require radical treatment due to PCa, fast development in surgical methods has been observed, especially minimally invasive methods, such as laparoscopic prostatectomy.

Continuous ambulatory peritoneal dialysis (CAPD) is a type of peritoneal dialysis that occurs in regular intervals during the day. It is an approachable form of renal replacement therapy for end-stage renal disease patients [3].

The aim of this work is to describe a case of a patient on CAPD, who underwent laparoscopic extraperitoneal radical prostatectomy (RP) due to incidentally diagnosed PCa during the pre-transplant evaluation. In this paper, we would like to present our experience with a PD patient treated with RP due to localized PCa, and contribute to the knowledge on the treatment of PCa patients with underlying health conditions. 

## 2. Case Report

A 50-year-old man with end-stage renal disease, who was on CAPD for 4 years, was diagnosed with prostate cancer cT2aN0M0. The patient underwent extraperitoneal laparoscopic radical prostatectomy. The diagnosis was established during the pre-transplant evaluation. 

The patient had preserved micturition and suffered from lower urinary tract symptoms, due to bladder outlet obstruction. He also suffered from erectile dysfunction. Uroflowmetry showed a pathologic flat and intermittent curve. Cystoscopy revealed a narrow bladder neck and a slightly enlarged median lobe of the prostate. The patient underwent a transurethral bladder neck incision, with resection of the median lobe. A Gleason score of six (3 + 3), indicating prostate cancer, was incidentally found during the histopathology specimen examination. The total PSA was 1.49 ng/mL, and a discreet difference in the texture of the left lobe was found during the digital rectal examination, qualified as cT2a. The MRI showed wedge-shaped hypointensities on T2-weighed scans in the peripheral zone of the prostate, defined as PIRADS2 lesions. The disease was assessed as low-risk prostate cancer and the patient was qualified for radical treatment. After presentation of the treatment options to the patient, he decided to undergo radical prostatectomy.

Meanwhile, the bladder neck stenosis recurred, manifesting as acute urinary retention. As the introduction of a catheter through the urethra failed, suprapubic cystostomy was performed. Moreover, due to a malfunction, the peritoneal dialysis catheter was revised and reimplanted two months before radical prostatectomy was performed. The peritoneal catheter was placed subcutaneously, with the exit site in a right lateral location, directed medially and downwardly, entering the peritoneal cavity in the hypogastrium (Figure 1a). As a preoperative antibiotic therapy, the patient received 1 g of cephazolin before surgery. LRP was performed using the extraperitoneal approach. The five trocars were inserted as shown in the picture below (Figure 1b).

An incision for insertion of the first trocar was made laterally to the umbilicus on the left, avoiding the subcutaneous route of the catheter (Figure 2).

The abdominal superficial fascia was carefully incised, and the extraperitoneal space was reached, behind the rectus sheath, by blunt finger dissection. The optical trocar was advanced, and a further working space was developed using a laparoscope. Next, the abdominal cavity was insufflated with CO_2_. The pressure for insufflation was lowered from standard 12 mmHg to 10 mmHg. To prevent intraperitoneal insufflation, three 5 mm trocars and one 10 mm trocar were inserted under direct vision. The peritoneum was shifted cranially and the peritoneal end of the CAPD tube was out of range for surgery. Adhesions were found in the hypogastrium, on the side of the previous catheter localization, preventing surgical entry into the true pelvis. After the adhesions were released, the Retzius’ space was reached and developed. The prostate gland with seminal vesicles and the ampullae of the vas deferens were approached, mobilized, and dissected. According to the guidelines for low-risk prostate cancer, lymphadenectomy was omitted. The specimen was entrapped in an endobag and removed through the camera port. After reapproximation of the remnant Denonvilliers’ fascia with the Rocco stitch [4], vesicourethral anastomosis was performed. A drain was inserted into the retroperitoneal space and all the incisions were closed with absorbable sutures. The fascial opening at the camera port site was sutured separately. During surgery, we used bipolar forceps as coagulation and laparoscopic scissors for dissection. The postoperative pathological examination revealed two foci of adenocarcinoma, with a GS of seven (3 + 4), in the left lobe of the prostate, with a diameter of up to 2 mm each. The surgical margins were clear (pT2Nx). In the perioperative period, hemodialysis was introduced and continued until CAPD was resumed. On the fourth postoperative day, the drain was removed. The bladder catheter was removed 11 days after prostatectomy.

In coordination with nephrologists, it was decided to resume dialysis 4 weeks after surgery. The patient did not require any additional antibiotic therapy after surgery.

At 6 months post-surgery, the patient regained almost full continence and only required one pad during the day.

## 3. Discussion

According to the pre-transplant guidelines of the Polish transplant center—“POLTRANSPLANT”—every patient who is being prepared for transplantation needs to be screened and treated for malignancies and functional disorders prior to surgery. Both active surveillance and focal therapy are not suitable, as the patient needs to be treated radically. Radical prostatectomy was chosen over radiotherapy; this was, firstly, because of the patient’s preference and, secondly, because the patient would reach nadir after RP in a shorter time compared to radiotherapy. Patients are eligible for kidney transplantation 2 years after RP [5].

There are few reports on radical prostatectomy in patients with CAPD, regardless of the technique. To the best of our knowledge, this is the first report to describe extraperitoneal laparoscopic RP in this group of patients.

Before the introduction of the laparoscopic technique, performing abdominal surgery on peritoneal dialysis patients was a challenge. To avoid the risk of perioperative complications, such as the leakage of dialysate fluid, peritonitis, and the development of intraabdominal adhesions, removal of the peritoneal catheter was considered mandatory [6]. Finally, PD resumption was an issue and failures occurred more frequently.

The introduction and continuous evolution of the laparoscopic technique involved a PD population. Its minimally invasive nature, associated with less peritoneal damage, enabled several surgical procedures, such as cholecystectomies, appendectomies, nephrectomies, and colectomies, to be performed without catheter removal and with early resumption of PD.

Radical prostatectomy is, alongside radiotherapy, the standard treatment option for patients with organ-confined prostate cancer [7]. The treatment modality decision depends on the perioperative risk assessment and the acceptance of potential complications specific to each approach. Radiotherapy is associated with an increased risk of developing late secondary malignancies, including bladder, colorectal and rectum cancers. The absolute additional risks over 10 years are about 1–4% [8]. As younger patients are still more likely to be started on PD [9], the late toxicity should be taken into consideration; moreover, younger patients are often more appropriate candidates for surgery.

There are the following two main approaches to laparoscopic radical prostatectomy: transperitoneal and extraperitoneal. Both may be performed with either conventional laparoscopic access or robot-assisted radical prostatectomy (RARP). The decision regarding the adopted technique depends on disease-specific factors and peritoneal dialysis implications. Patients with nodal involvement risk exceeding 5%, according to the Briganti score, require extended pelvic lymph node dissection, which may only be performed through transperitoneal access. This technique demands several peritoneal incisions and unavoidable peritoneal membrane stress, potentially leading to CAPD-related complications, such as leakage of dialysate fluid, wound dehiscence, incisional hernia, peritonitis, hemoperitoneum, and adhesions [10]. Kuribayashi et al. reported successful transperitoneal RALP with complete peritoneal repair and early reinstatement of PD [11].

The extraperitoneal approach allows the peritoneal membrane integrity to be preserved. Moreover, as we demonstrated, this access does not require CAPD catheter removal to insert the abdominal trocars properly. Taking the position of the CAPD tube into consideration, the position of the optical trocar was shifted laterally to the opposite side (Figure 2).

Laparo-endoscopic single-site surgery radical prostatectomy (LESS-RP) could be an option for prostate cancer patients with CAPD. In 2021, in Sortino, Giannoubilo et al. analyzed 520 patients who underwent RP with the use of a single-site incision technique. The analyses showed that the LESS-RP technique allows better aesthetic and psychological results, reduced postoperative pain, and a faster return to normal daily activity, with the same functional and oncological results as classic RP. It is a safe procedure, if performed by surgeons with adequate experience and skills [12]. However, this type of laparoscopic access is used by a limited number of institutions. In our opinion, conventional laparoscopic access provides the possibility to perform RLP in CAPD patients in a greater number of institutions. Surgeons, who use conventional laparoscopic access for RLP in everyday practice, will be able to perform the surgery in a shorter time, which is critical for these kinds of patients. The shorter time of surgery may lower the chance of disturbing the continuity of a peritoneum and lower the risk of infection.

The technical aspect, which could have critical potential to prevent further progress of the operation with extraperitoneal access, was the location of the intraperitoneal portion of the catheter. In the case of a typical end position in the middle hypogastrium, access into the true pelvis, without collision with the catheter, may not be possible. To avoid this obstacle, it was necessary to determine the catheter route at the stage of surgery planning. Cooperation with the surgeon who performed the tube insertion was necessary.

In our case, the patient returned to dialysis after 6 weeks, with a desirable outcome. Due to the small number of reported experiences in the literature, the optimal timing of postoperative peritoneal dialysis reestablishment is not clearly defined. If dialysis is started too early, potential complications, such as leakage of dialysate fluid, wound infection, or dehiscence, may occur [13]. Summarizing the analyzed available case reports, Mari et al. suggest a 4-week switch to hemodialysis as safe management for major and minor laparoscopic procedures. However, the reports on this issue differ throughout the literature. Szymanski et al. showed that nephrectomy with immediate postoperative peritoneal dialysis is a safe option for children with end-stage renal disease. They used retroperitoneoscopic access, and had to add temporary hemodialysis in only one case, because of a small peritoneal incision [14]. Kuribayashi reports a two-week CAPD interruption after transperitoneal prostatectomy, with tight suturing of the peritoneal membrane.

## 4. Conclusions

Radical prostatectomy in patients with CAPD may be performed safely through retroperitoneal access. This surgical approach allows peritoneum interruption to be avoided, but requires close multidisciplinary cooperation, including with surgeons and nephrologists, at the stage of operation planning.

## Figures and Tables

**Figure 1 life-12-00413-f001:**
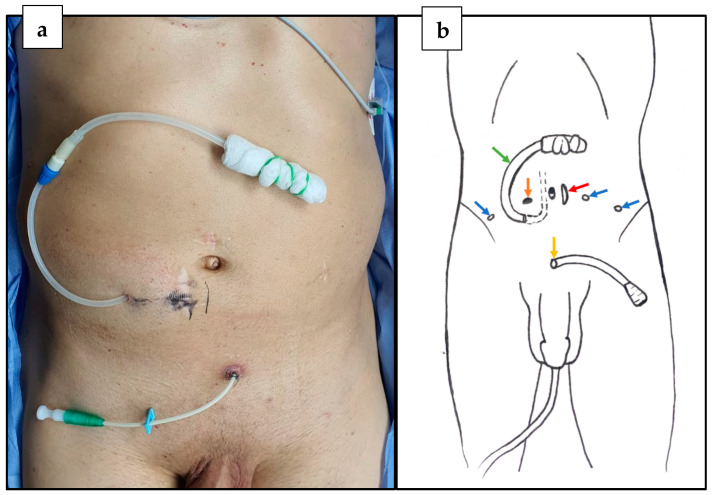
(**a**) Preoperative state. (**b**) Positioning of trocars during LRP vs. peritoneal catheter. Red arrow—12 mm optical trocar; blue arrows—5 mm trocars; orange arrow—10 mm trocar; green arrow—CAPD catheter (intraabdominal part marked with dashed lines); yellow arrow—cystostomy (graphic author: A.W.).

**Figure 2 life-12-00413-f002:**
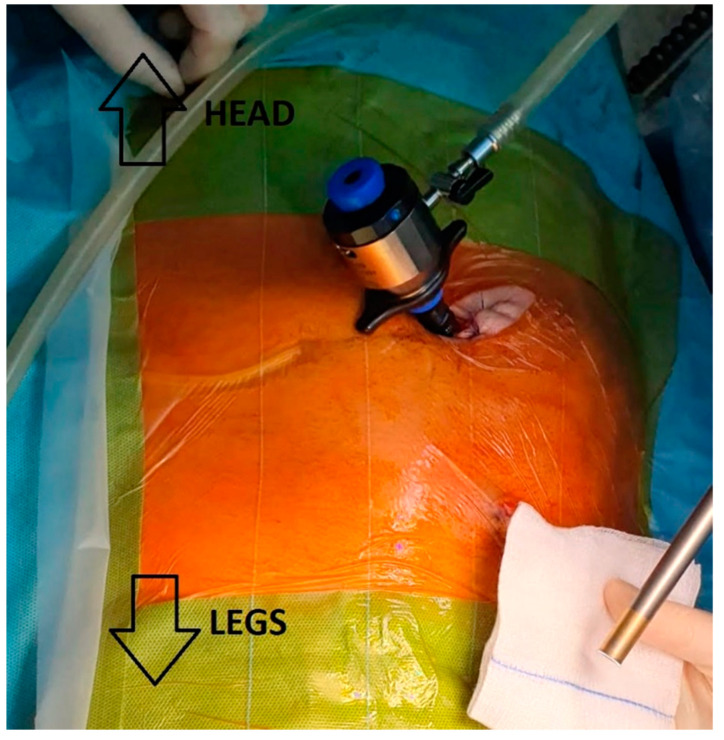
Positioning of the optical trocar. The trocar shifted laterally to the opposite side.

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
