# Peer review of "Extraperitoneal Laparoscopic Prostatectomy in a Prostate Cancer Patient Undergoing Continuous Ambulatory Peritoneal Dialysis"

_life, 2022, doi:10.3390/life12030413_

Round 1

Reviewer 1 Report

This manuscript presents a case report and from the point of scientific view, doesn’t present sufficient scientific soundness to be published in Life.  The case report describes a extraperitoneal laparoscopic prostatectomy (ELP) in a prostate cancer patient undergoing continuous ambulatory peritoneal dialysis (CAPD) and the application of both techniques doesn’t seem to have any novel findings. Authors may include a literature review prior case report description could increase the manuscript content. Although the manuscript is well-written and well presented the manuscript content does not seem new. The figure 3 is presented in a very low quality.

Author Response

Thank you for your opinion.

In this case major concern was an extraperitoneal approach of RP. We wanted to create the extraperitoneal space without disintegration of peritoneal lining. The most challenging technical aspects were the presence of CAPD catheter and cystostomy. In literature there are some publications that describe intraperitoneal RP with concomitant CAPD and descriptions of RP after kidney transplant. Best to our knowledge there is no other case report which describes extraperitoneal RP in patients with CAPD, before kidney transplant.

We did our best to improve the quality of Figure 3 however we are limited by the quality of the original photo taken during the surgery. We are very sorry about the poor quality, and we hope it is still readable thanks to our additional marking.

Reviewer 2 Report

Aim of the study was to describe a case of a patient on CAPD who underwent the laparoscopic extraperitoneal radical prostatectomy due to incidentally diagnosed PCa during the pre-transplant evaluation. 

Authors must to discuss why surgical treatment for a very low-risk Pca was performed. Nowadays, there are several non-surgical treatments which reported to be comparable with standard surgical therapy. Focal therapy, RT, active surveillance were different suitable options for this case, and even if the postoperative course was uneventful, it is not clear why patient was treated. This is the main concern of the study, that undermine overall quality of manuscript. 

Author Response

Thank you for your review.

The patient was diagnosed as low risk PCa due to the positive cT2a DRE examination, Gleason score <=6 and PSA level <10mg/dl.

He was qualified for radical treatment since according to pre-transplant guidelines of polish transplant center – „POLTANSPLANT”, patient needs to be treated radically before transplant. Both active surveillance and focal therapy were not suitable. Radical prostatectomy was chosen over radiotherapy, firstly because of the patient's preference and secondly because the patient reaches nadir after RP in a shorter time, comparing to radiotherapy. Patients are eligible for kidney transplantation in 2 years after RP.   

We added the comment in the discussion (row 117-123).

Reviewer 3 Report

The study described an innovative continuous ambulatory peritoneal dialysis (CAPD) surgical procedure in patients treated by retroperitoneal laparoscopic radical prostatectomy (RLRP). The reviewer highly recommends publication after authors consider the following minor revision if possible.

1) in introduction part, the author should discuss the relationship between CAPD and RLRP. Does that mean all the patients treated by RLPP will also be treated by CAPD? or Does this CAPD procedure apply to all patients with lower urinary tract symptoms due to bladder outlet obstruction no matter whether they undergo RLRP or not? The author should further clarify in what situation this CAPD should be applied and its unique advantage.

2) Figure 1 and Figure 2 can be combined to one figure in order to be easily compared and interpreted.

3) Figure 4. The author should integrate the Figure 4 into main text part of discussion part or case report part. Or else it will confuse the reader about the relationship of this Figure 4 with main text of this article.

Author Response

Thank you very much for your opinion.

1) Patient was undergoing CAPD for 4 years, prior to urological treatment, due to kidney failure. This technique was chosen independently to further urological conditions. Patient was undergoing hemodialysis only perioperatively and CAPD was restarted as soon as possible after surgery. CAPD catheter and adhesion after its placement were the main technical concerns as there was higher risk of peritoneal lining disturbance.

2) Figure 1 and 2 have been combined as you recommended. (row 21-77)

3) Figure 4 was integrated with the main text. (row 81-82).

Reviewer 4 Report

General comments

The authors reported the successful case of extraperitoneal LRP and its perioperative management in detail. This is a potentially difficult case to manage complications, and the report of the successful management without major complications will be useful for treatment selection in prostate cancer patients on CAPD.

Specific comments

  1. The timing of drain removal and bladder catheter removal was longer than usual. If CAPD-related complications were concerned, earlier removal or omitting of a drain was considered. How did you decide the timing of drain removal? Describe this.
  2. Why was preoperative one week chosen as the timing and duration of antimicrobial use? Regarding the reasoning behind this, previous reports and references should be introduced in the Discussion section.
  3. Active surveillance would be considered as a treatment option in addition to surgery and radiotherapy for organ-confined prostate cancer, especially in CAPD patients. Please include this treatment option in the Discussion section.
  4. One of the concerns with extraperitoneal LRP/RARP is intraoperative peritoneal injury. In the case of CAPD patients, in particular, is there any thought or preparation given to what to do in the event of peritoneal injury?

Author Response

Thank you for your opinion.

1. It was a standard timing of a drain and catheter removal in our department. The was no complications that influenced the timing.

2. Patient only received 1 dose of 1g cefazolin before surgery. Duration of antibiotic therapy was strictly planned according to nephrologists’ consultation and EAU recommendations.

3. We introduced changes in the discussion section regarding treatment options (row 117-123).

4. In case of peritoneal injury Kuribayashi et al. in their article: “Robot-Assisted Laparoscopic Prostatectomy in a Prostate Cancer Patient 232 Undergoing Continuous Ambulatory Peritoneal Dialysis” recommend tight suturing of the peritoneum and restarting CAPD after 2 weeks period.

Round 2

Reviewer 1 Report

Authors did not follow my recommendation and did no improve literature review. Authors reply was based only in justification regarding case nonvital. Therefore, I still think the manuscript is not suitable for publication since no modifications were done.

Author Response

Thank you for your review. 

We decided to remove Figure 3, due to its low quality and no major significance. 

Lack of literature in this topic does not allow us to carry out the complete literature review, therefore the analysis of existing publications is performed in the discussion section. 

We still strongly believe that this is a valuable case report which makes a contribution to existing literature.  

Reviewer 2 Report

no major comments

Author Response

Thank you for your answer.